# In Situ Nano-SiO_2_ Electrospun Polyethylene-Oxide-Based Nano-Fiber Composite Solid Polymer Electrolyte for High-Performance Lithium-Ion Batteries

**DOI:** 10.3390/nano13071294

**Published:** 2023-04-06

**Authors:** Luwei Shi, Longxing Zhang, Yanping Yang, Haipeng Zhang, Ruijie Yao, Caoquan Yuan, Shaobo Cheng

**Affiliations:** 1School of Materials Science and Engineering, Shanghai University of Engineering Science, Shanghai 201620, China; 18964409153@163.com (L.S.); 18234078321@163.com (L.Z.);; 2School of Chemistry and Chemical Engineering, Shanghai Jiao Tong University, Shanghai 200240, China

**Keywords:** all-solid-state lithium battery, solid-state polymer electrolyte, in situ SiO_2_ nanospheres, electrostatic spinning, electrochemical properties

## Abstract

Polyethylene oxide (PEO)-based composite polymer electrolytes (CPEs) containing in situ SiO_2_ fillers are prepared using an electrostatic spinning method at room temperature. Through the in situ hydrolysis of tetraethyl silicate (TEOS), the generated SiO_2_ nanospheres are uniformly dispersed in the PEO matrix to form a 3D ceramic network, which enhances the mechanical properties of the electrolyte as a reinforcing phase. The interaction between SiO_2_ nanospheres and PEO chains results in chemical bonding with a decrease in the crystallinity of the PEO matrix, as well as the complexation strength of PEO chains with lithium ions during the hydrolysis process. Meanwhile, the addition of SiO_2_ nanospheres can disturb the orderliness of PEO chain segments and further suppress the crystallization of the PEO matrix. Therefore, improved mechanical/electrochemical properties can be obtained in the as-spun electrolyte with the unique one-dimensional high-speed ion channels. The electrospun CPE with in situ SiO_2_ (10 wt%, ca. 45 nm) has a higher ionic conductivity of 1.03 × 10^−3^ S cm^−1^ than that of the mechanical blending one. Meanwhile, the upper limit of the electrochemical stability window is up to 5.5 V versus Li^+^/Li, and a lithium-ion migration number can be of up to 0.282 at room temperature. In addition, in situ SiO_2_ electrospun CPE achieves a tensile strength of 1.12 MPa, elongation at the break of 488.1%, and it has an excellent plasticity. All in all, it is expected that the electrospun CPE prepared in this study is a promising one for application in an all-solid-state lithium-ion battery (LIB) with a high energy density, long life cycle, and high safety.

## 1. Introduction

With the rapid development of electric vehicles, more attention has been paid to increasing the high safety performance, long life cycle, and charging rate of lithium-ion batteries (LIBs). Although the traditional organic liquid LIBs have an excellent electrochemical performance, LIB safety accidents have occurred frequently due to the lithium dendrite growth inevitably causing an explosion, and the fact that flammability and leakage of their liquid electrolytes leads to thermal runaway [1,2,3]. As a stark contrast, solid-state polymer electrolytes are expected to be the key to solve the existing problems faced by the organic liquid electrolyte [4,5,6,7,8]. 

Solid polymer electrolytes (SPEs) with PEO as the matrix have received increasing attention due to their excellent safety properties, ease of processing, and their complexation with lithium ions [9,10,11]. However, PEO is a semi-crystalline polymer with a small percentage of amorphous region available for lithium-ion conduction, which leads to low room temperature ionic conductivity and a poor electrochemical performance. In addition, the large application with PEO-based SPE in LIBs has been also severely plagued by the trade-off between ionic conductivity and mechanical properties [12,13,14]. To address the above problems, the addition of inorganic nanofillers such as Al_2_O_3_ [15,16], ZrO_2_ [17], and SiO_2_ [9,18] into the PEO matrix have been confirmed as a very useful strategy to prepare composite polymer electrolytes (CPEs) with a superior performance [19,20,21]. The incorporation of inert fillers can disturb the ordered arrangement of macromolecular chains which reduces the crystallinity of the polymer matrix. Therefore, an increase in the area of the amorphous region can be obtained, which in turn improves the ionic conductivity of CPEs [21,22,23,24,25,26]. Furthermore, the Lewis-acid-based pair interaction of ceramics fillers–lithium salt can significantly encourage dissociation of lithium salt and stabilizing the anions. Meanwhile, the mechanical properties of CPEs are greatly improved due to the addition of these reinforcing phases. Thus, motivated by these advantages, the CPE composed of polymer, Li salt, and filler not only has the advantages of the flexibility and machinability of SPEs, but it can also achieve a high ionic conductivity. However, the ceramic-polymer-based CPE by the direct addition of nanoparticles to the PEO matrix still experiences the following challenges: (1) Li^+^ conduction in the CPE may be hindered by the interfacial resistance between the filler and the polymer matrix, as well as the grain boundary between the filler particles, resulting in a high energy barrier and low cation migration number of interface ion transport, and (2) nano-sized fillers tend to agglomerate and destroy the seepage network in the CPE, leading to great challenges in further improving the ionic conductivity.

As an advanced preparation method of PEO-based SPEs, electrostatic spinning is expected to achieve a unique 3D network structure of nanofiber composite electrolyte [27]. This structure not only provides a one-dimensional ion transport channel to accelerate the conduction of lithium ions [28,29,30,31], it also improves the mechanical strength of the electrospun electrolytes [32,33].

Herein, based on the advantages of electrostatic spinning technology, a novel electrospun CPE with the in situ generation of inorganic nanofillers was prepared, whereby the unique 3D network structure of as-spun CPE was obtained. The SiO_2_ nanospheres are uniformly dispersed in the PEO matrix, as the in situ hydrolysis of tetraethyl silicate (TEOS) occurs in the PEO solution. The SiO_2_ nanospheres can be chemically and physically interacted with PEO chain segments, thus resulting in new chemical bonds and strong interactions [34]. The crystallization of PEO chain segments is then effectively inhibited, and this results in an increase in the proportion of amorphous regions. In addition, the spinning solution is ejected at a high speed and rapidly formed under high pressure, which also plays a role in restraining the crystallization of PEO chain segments [32]. Therefore, these factors result in the highest ionic conductivity of 1.03 × 10^−3^ S cm^−1^, an upper limit of electrochemical stability window of up to 5.5 V, and a lithium-ion transference number of up to 0.282 for the electrospun nanofiber electrolyte.

## 2. Experimental Methods

### 2.1. Materials

Anhydrous ethanol (99.7%) was obtained from Titan Chemical Co., Ltd., Hangzhou, China. Aqueous ammonia (25–28%) was purchased from Sinopharm Chemical Reagent Co., Ltd., Shanghai, China. Ethyl orthosilicate (TEOS, 99%), acetonitrile (99.9%), polyethylene oxide (PEO, M_v_ ~ 1,000,000), and lithium bis(trifluoromethanesulfonyl)imide (LITFSI, 99.9%) were all purchased from Aladdin Biochemical Technology Co., Ltd., Shanghai, China. All chemical reagents were used without any purification process.

### 2.2. Preparation of the Eletrospun CPEs

(1)Preparation of the seed solution with the monodisperse of 12 nm SiO_2_ nanospheres: 30 mL of deionized water was added in a 100 mL beaker. The pH of the deionized water was adjusted to 10.8 with ammonia–water. In total, 2.1 g of TEOS was added to the ammonia–water by drop until we generated a seed solution of monodisperse 12 nm SiO_2_ nanoparticles at 60 °C [34].(2)Preparation of the monodisperse of 45 nm SiO_2_ nanospheres: 4 mL of aqueous ammonia was mixed with 29 mL of anhydrous ethanol and stirred well to make the ammonia precursor solution. The first SiO_2_ nanosphere solution was prepared by mixing 0.6 g of seed solution with 30 mL of deionized water by mechanical stirring. Then 10 mL of ammonia precursor solution and 0.2 mL of first SiO_2_ nanosphere solution was added to 25 mL of anhydrous ethanol to obtain a homogeneous mixture. After that, 0.347 g of TEOS was added dropwise to the resulting solution by continuous stirring, and then the second SiO_2_ nanosphere (monodisperse, ca. 45 nm,) solution was obtained.(3)Preparation of mechanically doped spinning solutions: the PEO solution was made by dissolving 0.6994 g of PEO with 25 mL of acetonitrile. The PEO/LiTFSI spinning solution was prepared by adding LiTFSI with PEO in the molar ratio of 1:16 to the PEO solution. Then, monodisperse 45 nm SiO_2_ nanospheres were evenly dispersed in PEO/LiTFSI solution under ultrasonic stirring. The obtained solution subsequently underwent evaporation at 50 °C for 6 h in a vacuum oven to produce a spinning solution mechanically doped with 45 nm SiO_2_ nanospheres (PEO/LiTFSI/MD-10%SiO_2_); similarly, a spinning solution (PEO/LiTFSI/MD-S10%SiO_2_) with mechanically doped 12 nm SiO_2_ nanospheres was produced.(4)Preparation of in-situ-generated spinning solution: PEO solution was added to 25 mL of anhydrous ethanol, then 10 mL of ammonia precursor solution and 0.2 mL of the first SiO_2_ nanosphere solution was added and stirred at room temperature. After complete dissolution, 0.347 g of TEOS was added drop-by-drop to the above solution, after which the stirring was continued to obtain the in situ SiO_2_ (45 nm)-PEO solution. The spinning solution (PEO/LiTFSI/ISF-10%SiO_2_) containing in-situ-generated 45 nm SiO_2_ nanospheres was obtained by adding the dried and dehydrated LITFSI to the in situ SiO_2_ (45 nm)-PEO solution (the molar ratio of PEO and LiTFSI is 16:1), and then this was evaporated in a vacuum oven at 50 °C for 6 h. Similarly, by changing the weight percentage of in situ SiO_2_ nanospheres in the system, the electrospinning solution of PEO/LiTFSI/ISF-5%SiO_2_ and PEO/LiTFSI/ISF-15%SiO_2_ was also prepared. The properties of SiO_2_ in the composite solid polymer electrolyte are shown in Table 1.

(5)Preparation of the mechanically doped and in-situ-generated electrospun CPEs: the spinning solution was electrostatically spun under a high electrostatic pressure. By adjusting the technical parameters of electrospinning, the electrospun CPEs with a thickness between 50 and 200 μm was achieved. The as-spun electrolytes were then punched into 19 mm diameter discs and vacuum-dried for 24 h at 45 °C. Finally, the electrolytes were transferred into the glove box for a later use.

### 2.3. Structure Characterization

Scanning electron microscopy (FE-SEM, Sigma 300, Zeiss, Oberkochen, Germany) was used to observe the morphology, nanoparticle dispersion, and particle size of the fibrous films with an electron beam of 3 kV. The elemental distribution of the electrospun CPEs was studied by energy dispersive X-ray spectroscopy (EDS) with an electron beam of 15 kV. The phase structure and crystallinity of pristine PEO and electrospun CPEs were observed by an X-ray diffractometer (XRD, D2 Phaser, Bruker, Mannheim, Germany) with a scanning angle range *2θ* of 5–90° and a scanning speed of 2°/min using CuKα radiation with λ = 1.5418 Å. The interaction between in situ SiO_2_ nanospheres and PEO chains was characterized by FTIR spectra (Scientific Nicolet iS20, Shanghai, China). The mechanical properties of the as-spun CPEs were recorded using a Universal Material Testing Machine (LDW-5, Shanghai Songton Instrument Manufacturing Co., Ltd., Shanghai, China).

The crystallinity ($\chi_{c}$) of the as-spun CPEs was determined using differential scanning calorimetry (DSC, DSC 200 F3, Netzsch, Selb, Germany) in the temperature range −80–180 °C with a ramp rate of 10 °C/min. *χ_c_* was calculated using the equation:(1)$$\chi_{c}=\frac{{\Delta H}_{m}}{{\Delta H}_{m}^{*}}\times 100\%$$
where ${\Delta H}_{m}$ is the melting enthalpy of CPEs and ${\Delta H}_{m}^{*}$ is the melting enthalpy of PEO in the fully crystallized state with a value of 177.8 J g^−1^.

### 2.4. Electrochemical Characterization 

The electrospun CPE with a diameter of 19 mm was assembled into a stainless-steel sheet (SS)/electrospun CPE/SS sandwich structure cell. The ionic conductivity (σ, Equation (2)) for the as-spun CPEs was tested using an EIS with a frequency range of 10^5^~10^−2^ Hz under an electrochemical workstation (CHI 760E, Shanghai C&H Instruments Co., Ltd., Shanghai, China) at 60 °C.
(2)$$\sigma =\frac{d}{R_{b}S}$$
where *σ* is the ionic conductivity, *d* is the thickness of CPE, *R_b_* is the bulk resistance, and *S* is the area of the SS (1.96 cm^2^).

The electrochemical stability window of Li/electrospun CPE/SS was measured using a linear sweep voltammetry (LSV) method with the voltage range of 0–6 V at 60 °C.

A combination of the AC impedance method and DC polarization method was used to obtain AC impedance profiles and polarization curves of Li/electrospun CPE/Li symmetric cells at 20 °C. The lithium-ion migration number was calculated as follows:(3)$$t_{{Li}^{+}}=\frac{I_{s}(\Delta V-I_{0}R_{0})}{I_{0}(\Delta V-I_{s}R_{s})}$$
where $t_{{Li}^{+}}$ is the lithium-ion migration number. *R*_0_ and *R_S_* denote the impedance values before and after polarization, respectively, Δ*V* is the magnitude of the applied polarization potential (10 mV), and *I*_0_ and *I_S_* are the initial and steady-state currents. 

## 3. Results and Discussion

Figure 1 shows the preparation process and multiple conduction modes of lithium ions of in-situ-generated electrospun CPE, respectively. The spinning solution containing in situ SiO_2_ nanospheres is electrostatically spun under a high pressure and, subsequently, the fibers are interleaved and deposited on the receiver plate to finally form an electrospun electrolyte with a 3D network structure. Therefore, the Li^+^ ions of in-situ-synthesized CPE can be transported by the complexation interaction between lithium ions and C-O-C bond from the PEO matrix. Additionally, the electrospun technology offers the opportunity to conduct lithium ions rapidly in the interfacial layer channel between the SiO_2_ nanospheres and the PEO substrate [35]. Furthermore, the average size of SiO_2_ nanospheres can well be controlled to ca. 45 nm because of the in situ generation technique used (Figure 2e). Homogeneous nano-SiO_2_ with a large specific surface area can increase the effective surface area of transmission paths for lithium-ion. The combination of these conduction methods results in a battery with a high ionic conductivity and excellent electrochemical performance [34].

For the abovementioned multiple conduction modes of lithium ions, all of them are based on the low crystallinity of the PEO matrix. Compared with a simple mechanical doping method, the in situ generation and re-growth technology used in this study successfully dispersed SiO_2_ nanospheres uniformly into polymer hosts. At the same time, the -OH at the end of the PEO chain can also chemically bond with the -OH on the surface of SiO_2_ under hydrolysis conditions. These two interaction mechanisms result in a substantial reduction in the crystallinity of the PEO matrix by disturbing the ordered arrangement of the macromolecular chain segments [34].

### 3.1. Microstructure and Morphology of the Electrospun SPEs

Figure 2a–e shows the SEM images of SPEs with and without in situ SiO_2_ nanospheres prepared by the electrostatic spinning process. It can be seen that the electrostatically spun nanofibers were deposited interleaved to form a 3D network structure. Among them, the average diameter distribution of the fibers is more uniform and slenderer compared to that of the SPEs without SiO_2_ nanospheres, regardless of whether they contain mechanically doped SiO_2_ nanospheres or in-situ-generated SiO_2_ nanospheres. This phenomenon is due to the fact that the Si element in the electrospun electrolyte increases the electrical conductivity of the spinning solution, so as to improve the charge density of the jet, and then promote the jet to split into nanofibers [36]. As shown in Figure 2d,e, it can be clearly seen that the 45 nm SiO_2_ nanospheres added by magnetic stirring are heavily agglomerated, while the 45 nm SiO_2_ nanospheres of the in situ generation can be uniformly dispersed in the PEO matrix. In addition, the interface is fuzzy with no significant pinholes or interfacial cracks between in situ SiO_2_ nanospheres and the PEO matrix on the fiber surface (Figure 2e). These results confirmed that there is good interaction and interfacial compatibility in the in situ silica and polymer matrix [37].

EDS images are used to further characterize these morphological features (Figure 2g–i). The bright dots indicates that the S and F of LiTFSI are evenly impregnated into both PEO matrixes, indicating that all electrospun SPEs are successfully prepared with electrospinning technology. In stark contrast, the distribution of the Si element on the surface of in-situ-generated electrospun CPE is more uniform than that on the mechanically doped one. It may be that SiO_2_ nanospheres exhibit severe agglomeration when mechanically doped into the PEO matrix, whereas the agglomeration of in-situ-generated SiO_2_ nanospheres is significantly improved, in line with the earlier SEM analysis. 

FTIR spectra are used to verify the interaction between in situ SiO_2_ nanospheres and PEO chain segments. The spectra of electrospun SPEs with and without SiO_2_ nanospheres are presented in Figure 3a,b. According to literature reports [38,39,40], the C-O-C stretching spectrum of pristine PEO is located at 1114 cm^−1^, while the characteristic peak in that of PEO/LiTFSI decreases to 1094 cm^−1^ with the addition of LiTFSI, indicating that Li^+^ complexation with C-O-C occurs (Figure 3a). The characteristic absorption peak of SiO_2_ nanoparticles is located at 467 cm^−1^ [41], and the peaks are shifted to 459 cm^−1^ (PEO/LiTFSI/MD-S10%SiO_2_) and 455 cm^−1^ (PEO/LiTFSI/ISF-10%SiO_2_), respectively (Figure 3b). This fact may be ascribed to disturbing the ordered arrangement of PEO chains which occurs due to the nano-additives, as well as a strong interaction between in-situ-generated SiO_2_ nanospheres with PEO chains. In particular, it is worth noting that chemical bonding occurs and stabilizes the vibration of the Si-O-Si chain segment, which is well consistent with the report of Dingchang Lin [34].

The effect of LiTFSI and SiO_2_ nanospheres on the crystallinity of PEO is investigated by XRD tests. As can be seen from Figure 3c, PEO is a semi-crystalline polymer with crystalline characteristic peaks located at *2θ* = 19.2° (120) and *2θ* = 23.2° (112) [42]. With the addition of LiTFSI, 2 new peaks appeared between 65° and 80°, which are the characteristic peaks of LiTFSI. Additionally, the intensity of the *2θ* = 23.2° (112) peak decreases because TFSI^-^ can disrupt the orderliness of PEO chains, resulting in the low crystallinity of the PEO matrix. Moreover, the peak intensity has further been reduced due to the addition of 45 nm SiO_2_ nanospheres both in mechanical doping and the in situ generation of electrospun CPE. Interestingly, the modification effect of in situ nano-SiO_2_ on the PEO matrix is more obvious with a lower crystallinity, which is due to the fact that in situ SiO_2_ nanospheres can be uniformly distributed in the PEO matrix.

Similarity, the change in the thermal properties can also confirm the effect of LiTFSI or in situ SiO_2_ nanospheres on the chain segment mobility of the PEO matrix. The DSC curves of pristine PEO and PEO/LiTFSI/ISF-10%SiO_2_ are presented in Figure 3d. The results show that there is only one endothermic peak in the range of 40–80 °C, which belongs to its melting temperature (*T_m_*). As shown in Table 2, the values of *T_m_*, Δ*H_m_*, and *χ_c_* for PEO/LiTFSI/ISF-10%SiO_2_ are 57.59 °C, −44.62 J g^−1^, and 25.10%. By comparison, the corresponding thermal properties of the PEO/LiTFSI electrolyte prepared by Zhang Yi et al. are 56.51 °C (*T_m_*), −53.35 J g^−1^ (Δ*H_m_*), and 30.01% (*χ_c_*), respectively [43]. Obviously, a decrease in T_m_, ΔH_m_, and *χ_c_* occurs with the successive addition of LiTFSI and in-situ-generated SiO_2_ nanospheres. These results may be due to the chemical bonding and mechanical wrapping that occurs during the in situ hydrolysis process, which reduces the crystallinity of PEO/LiTFSI/ISF-10%SiO_2_ and improves the conductivity of Li^+^. However, the Tm of PEO/LiTFSI/ISF-10%SiO_2_ is higher than that of PEO/LiTFSI, indicating that the in situ nano-SiO_2_ also has a certain stabilizing effect on the PEO chain segments.

### 3.2. Mechanical Properties of As-Spun SPEs

In all-solid-state lithium batteries, the SPE serves to prevent physical contact between the positive and negative electrodes, whereby allowing a free transfer of Li^+^ in the electrolyte. Therefore, the improved mechanical performance of the SPE is significant for restraining the formation of lithium dendrites and achieving the high cycle stability of lithium metal batteries. Figure 4 shows the optical photographs under different mechanical forces and stress–strain curves of the electrospun SPEs samples. It can be found that PEO/LiTFSI/ISF-10%SiO_2_ has a smooth surface and excellent flexibility, which greatly facilitates the assembly of the cell. In addition, the tensile strength of the electrolyte is 0.13 MPa, with a % elongation at break of 340.2 when only PEO and LiTFSI are presented (Figure 4). With the addition of SiO_2_ nanoparticles, the tensile strength and elongation at the break of the as-spun electrolyte are greatly improved. This is due to the formation of a ceramic network, which serves as the PEO matrix’s reinforcing phase and increases the mechanical strength of the electrolyte when SiO_2_ nanoparticles are dispersed throughout it [25]. Furthermore, the tensile strength (1.12 MPa) and elongation at the break (488.1%) of PEO/LiTFSI/ISF-10%SiO_2_ are higher than those of PEO/LiTFSI/MD-10%SiO_2_. It may be ascribed to a more uniform structure caused by in-situ-synthesized SiO_2_ nanoparticles in the PEO matrix. Additionally, the PEO chain interacts with SiO_2_ nanoparticles, i.e., chemically bonded and mechanical encapsulation, thereby playing a role in locally pinning a PEO chain [34].

### 3.3. Electrochemical Properties of the Electrospun SPEs

Table 3 shows the influence mechanism of the nano-SiO_2_ content and adding methods, namely, mechanical doping and in situ generation, on the ionic conductivity of the as-spun SPEs. The corresponding impedance spectroscopy is shown in Figure 5a. As can be seen from Figure 5a and Table 3, an increase in the ion conductivity of the as-spun CPEs can be obtained with the addition of nano-silica into pristine PEO/LiTFSI. This result may be due to the uniform dispersion of nano-silica into the as-spun CPEs, which promotes the adsorption of the anions of the lithium salt by Lewis acid, thus effectively increasing the ionic conductivity. In addition, the impedance value of the electrospun CPE prepared by the in situ formation approach (PEO/LiTFSI/ISF-10%SiO_2_) is lower than that of mechanically doped CPE (PEO/LiTFSI/MD-10%SiO_2_) under the same particle size and content conditions of silica. Hence, the corresponding ionic conductivity of the former is twice as high as that of the latter. This phenomenon may be explained by the distribution of SiO_2_ nanospheres in PEO/LiTFSI/ISF-10%SiO_2_, which is more uniform than that of PEO/LiTFSI/MD-10%SiO_2_, and there is a strong interaction between in situ SiO_2_ nanospheres and PEO chain segments. In the presence of these combined effects, in situ SiO_2_ nanoparticles can more potently reduce crystallization and boost the fraction of the amorphous region of PEO matrix, as well as expand the active zone of the Lewis acid–base interaction. It is worth noting that the ionic conductivity of in-situ-generated CPEs firstly increases and then decreases with the increase in the content of SiO_2_ nanospheres. It indicates that in situ SiO_2_ nanospheres play a positive role in the modification of the PEO matrix. However, in situ SiO_2_ nanospheres may be seriously driven by thermodynamics with a content at 15 wt%, which plays a negative role in hindering the migration of lithium ions and worsening the modification effect [36].

Linear sweep voltammetry (LSV) is used to evaluate the electrochemical stability window (Figure 5b). It can be noted that the electrospun SPEs without SiO_2_ nanospheres show an obvious oxidation current at 4.8 V versus Li/Li^+^, and the SPEs with mechanically doped SiO_2_ nanospheres show an obvious oxidation current at 5.0 V versus Li/Li^+^. Interestingly, the LSV curve of the as-spun SPEs containing in situ SiO_2_ nanospheres shows that the oxidation occurred when versus Li/Li^+^ is up to 5.5 V. This result may be caused by a new chemical bond with the PEO chain during the in situ hydrolysis of TEOS as abovementioned. The new bond may interact with LiTFSI, thus promoting the decomposition and stability of the anion. Hence, the wide electrochemical stabilization window allows the as-spun CPEs to be used in a wide range of lithium storage systems, including high-voltage systems.

The lithium-ion transfer number ($t_{{Li}^{+}}$) is also an important factor in the performance of SPEs, as low $t_{{Li}^{+}}$ increases the polarisation of the electrode and reduces the performance of the cell. $t_{{Li}^{+}}$ of Li|PEO/LiTFSI/ISF-10%SiO_2_|Li symmetric cell is determined using the method proposed by Vincent and Evans, as shown in Figure 5c,d. The corresponding value is 0.282 at 20 °C (Table 4), which is higher than that of PEO/LiTFSI SPEs (0.141, 80 °C) without SiO_2_ nanospheres [43]. Corresponding to the results of EIS, this phenomenon can be attributed to in-situ-generated SiO_2_, which act as the physical crosslinking center, changing the structural state of the polymer chain segment, and providing more paths for Li^+^ transmission. In addition, the CPE prepared by the electrostatic spinning process has a unique 3D network structure, which can provide fast 1D channels of ion transmission for the conduction of lithium ions [28,29,30,31]. At the same time, due to the equilibrium relationship of the Lewis acid–base reaction, SiO_2_ and TFSI^-^ groups compete in binding Li^+^ [44], which promotes the dissociation of lithium salt and makes the transport of lithium ions easier [45,46,47].

## 4. Conclusions

A composite solid polymer electrolyte (CPE) containing in situ SiO_2_ nanospheres (ca. 45 nm, 10 wt%) has been prepared by electrostatic spinning. The interaction between in situ SiO_2_ nanospheres and PEO chains, i.e., chemical bonding and mechanical wrapping, can reduce the crystallinity of the PEO matrix to 25.1%. Meanwhile, the complexation between the Li^+^ and C-O-C of PEO chains is obviously weakened by in-situ-synthesized SiO_2_ nanospheres, thus promoting the easier migration of lithium ions. Therefore, the electrospun CPE containing 10 wt% 45 nm in situ SiO_2_ nanospheres has the highest ionic conductivity (1.03 × 10^−3^ S cm^−1^ at 60 °C), and its electrochemical stability window widened to 5.5 V, indicating that this system can be used in a wide range of lithium storage systems, even high-voltage systems. In conclusion, the unique 3D network structure and excellent electrochemical properties of the in-situ-generated electrospun CPE will provide a new strategy for the long-life operation of all-solid-state batteries.

## Figures and Tables

**Figure 1 nanomaterials-13-01294-f001:**
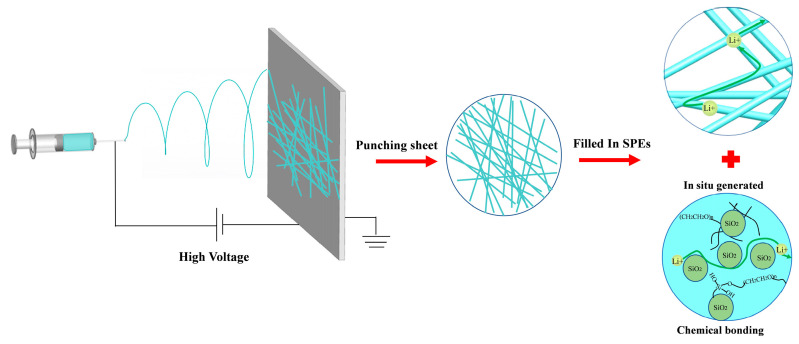
Schematic diagram for preparation of in-situ-generated electrospun CPE, and the interaction mechanism of in-situ-generated SiO_2_ on PEO chains.

**Figure 2 nanomaterials-13-01294-f002:**
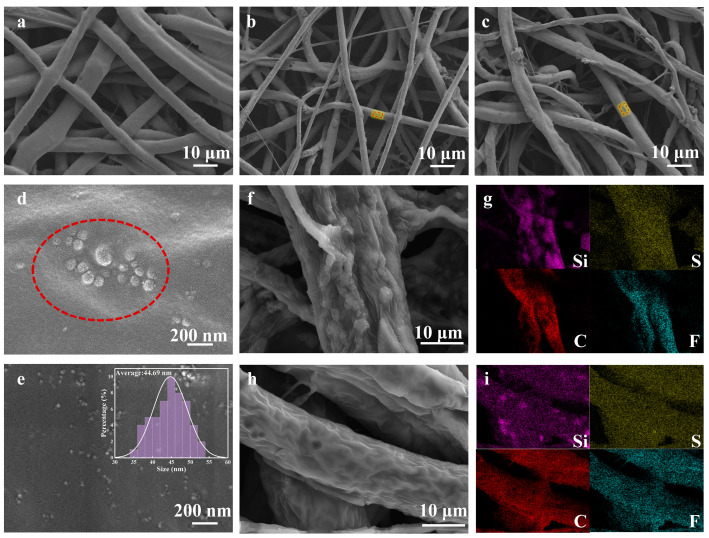
SEM images of as-spun SPEs with (**a**) no SiO_2_, (**b**) 45nm 10 wt% SiO_2_ of mechanical doping, (**c**) 45nm 10 wt% SiO_2_ of in situ formation, (**d**) the partial enlarged view in box of (**b**), and (**e**) the partial enlarged view in box of (**c**). EDS mapping of Si, C, F, and S elements in 45 nm 10 wt% SiO_2_ of mechanical doping (**f**,**g**) and 45nm 10 wt% SiO_2_ of in situ formation (**h**,**i**).

**Figure 3 nanomaterials-13-01294-f003:**
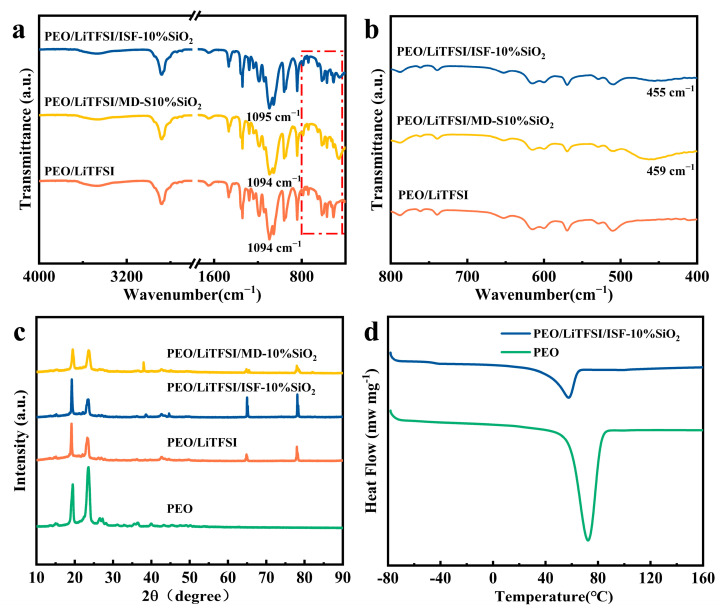
(**a**) FTIR spectra of as-spun SPEs. (**b**) Local magnification of 400–800 cm^−1^ in (**a**). (**c**) X-ray patterns and (**d**) DSC curves of as-spun SPEs.

**Figure 4 nanomaterials-13-01294-f004:**
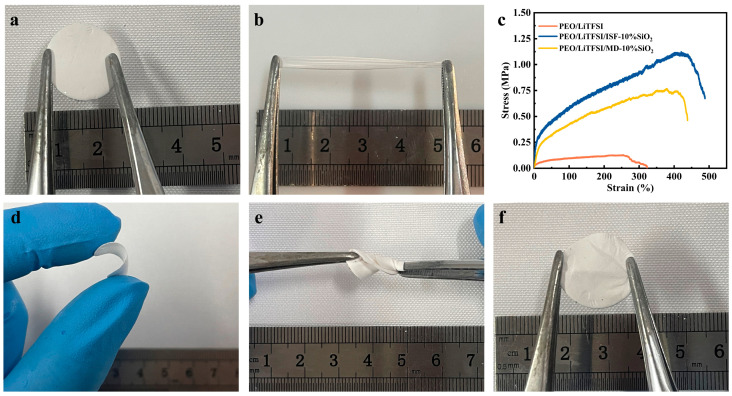
(**a**) Initial and (**b**) stretching appearance of PEO/LiTFSI/ISF-10%SiO_2_. (**c**) Stress–strain curves of the electrospun SPEs. (**d**) Bending, (**e**) torsion, and (**f**) recovery appearance after torsion of PEO/LiTFSI/ISF-10%SiO_2_.

**Figure 5 nanomaterials-13-01294-f005:**
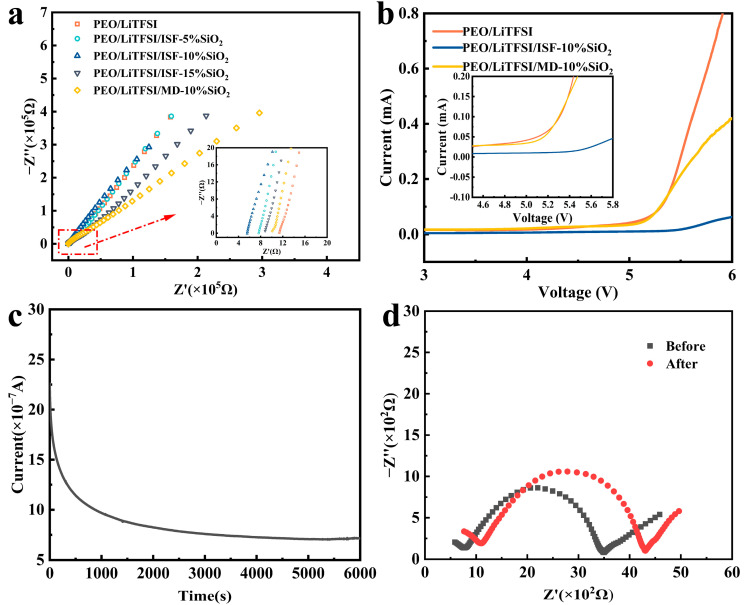
(**a**) Nyquist plots and (**b**) linear sweep voltammetry of the as-spun SPEs at 60 °C, (**c**) DC polarization, and (**d**) EIS curves of PEO/LiTFSI/ISF-10%SiO_2_ at 20 °C.

**Table 1 nanomaterials-13-01294-t001:** A summary of the mass loading and adding ways of SiO_2_ for the electrospun CPEs.

Samples	Diameter/nm	Content/wt%	Adding Ways
**PEO/LiTFSI/ISF-5%SiO_2_**	45	5	In situ generated
**PEO/LiTFSI/ISF-10%SiO_2_**	45	10	In situ generated
**PEO/LiTFSI/ISF-15%SiO_2_**	45	15	In situ generated
**PEO/LiTFSI/MD-10%SiO_2_**	45	10	Mechanical doping
**PEO/LiTFSI/MD-** **S10%SiO_2_**	12	10	Mechanical doping

**Table 2 nanomaterials-13-01294-t002:** The thermal properties value of pristine PEO and PEO/LiTFSI/ISF-10%SiO_2_.

Samples	*T_m_*/°C	Δ*H_m_*/J g^−1^	$\chi_{c}$/%
**PEO**	72.36	−170.10	95.67
**PEO/LiTFSI/ISF-10%SiO_2_**	57.59	−44.62	25.10

**Table 3 nanomaterials-13-01294-t003:** The values of the ionic conductivity(σ) for the electrospun SPEs.

Samples	*d*/cm	*R_b_*/Ω	*σ*/S cm^−1^
**PEO/LiTFSI**	1.14 × 10^−2^	11.370	5.1 × 10^−4^
**PEO/LiTFSI/ISF-5%SiO_2_**	1.29 × 10^−2^	7.640	8.6 × 10^−4^
**PEO/LiTFSI/ISF-10%SiO_2_**	1.12 × 10^−2^	5.571	1.03 × 10^−3^
**PEO/LiTFSI/ISF-15%SiO_2_**	1.29 × 10^−2^	8.814	7.5 × 10^−4^
**PEO/LiTFSI/MD-10%SiO_2_**	1.07 × 10^−2^	10.090	5.4 × 10^−4^

**Table 4 nanomaterials-13-01294-t004:** The lithium-ion transference number ($t_{{Li}^{+}}$) of PEO/LiTFSI/ISF-10%SiO_2_.

Samples	** *I* _0_ ** **/μA**	** *I_S_* ** **/μA**	** *R* _0_ ** **/Ω**	** *R_S_* ** **/Ω**	**Δ** ** *V* ** **/mV**	** *T* ** **/°C**	$t_{{Li}^{+}}$
**PEO/LiTFSI/ISF-10%SiO_2_**	2.26	0.71	760.6	1094	10	20	0.282

## Data Availability

Not applicable.

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
