# Peer review of "In Situ Nano-SiO2 Electrospun Polyethylene-Oxide-Based Nano-Fiber Composite Solid Polymer Electrolyte for High-Performance Lithium-Ion Batteries"

_nanomaterials, 2023, doi:10.3390/nano13071294_

Round 1

Reviewer 1 Report

The authors prepare by electrospinning and characterize a composite polymer electrolyte (CPE) based on Polyethylene oxide (PEO) and LiTFSI embedding SiO2 nanoparticles generated in situ. The characterization techniques demonstrate (i) homogeneous distribution of SiO2 nanospheres (about 45 nm diameter) and LiTFSI in the PEO matrix; (ii) reduced crystallinity of the PEO matrix and weakening of the Li+/PEO interactions; (iii) enhanced ionic conductivity and wide electrochemical stability window; (iv) improved mechanical properties, compared to the mechanically doped SiO2 nanospheres PEO-LiTFSI and to LiTFSI/PEO.

The subject is appropriate for publication in “Nanomaterials”. The investigated CPE seems promising to develop LIBs with long-cycle life and high charging rate. These are key aspects in the development of electric vehicles. Anyway, I think the paper should be revised by addressing the following points (major revision).

1)      A number of samples were synthesized by changing the strategy to embed the SiO2 nanoparticles into PEO (mechanically doped or in-situ), the SiO2 amount (5, 10 or 15 wt%) and the SiO2 nanoparticles diameter (12 or 45 nm). I think useful for the reader to add a Table summarizing the samples features and the codes used in the text in paragraph 2.2.

2)      Paragraph 2.2. Lines 111-112. The authors report a spinning solution (PEO/LiTFSI/MD-S10 wt% SiO2) with mechanically doped 12 nm SiO2 nanospheres was produced, but I did not find comments on this sample in the paper.

3)      Figure 1 caption: it refers to the in-situ process, while in the figure also mechanical doping is reported.

4)      Page 7, lines 237-248. While it is reasonable the PEO peaks intensities decrease by adding LiTFSI and SiO2, confirming lower crystallinity of the matrix, it is not so obvious to attribute the (120) peak intensity increase to the SiO2 –PEO interaction (lines 246-248).

5)      The following sentences do not sound, and the English language should be improved:

Lines 214-218

Lines 275-276

6)      Typos:

Line 26: change “prepares” into “prepared”

Line 104: change “was obtain” into “was obtained”

Line 134: change “was study” into “was studied”

Line 138: change “was charactered” into “was characterized”

Line 202: change “electrospuun” into “electrospun”

Figure 3 caption: change “400-600” into “400-800”

Line 239: change “locates” into “located”

Lines 285-286: change “It is may be ascribed to” into “It may be ascribed to”

Author Response

Dear Reviewer:

    On behalf of my co-authors, we thank you very much for giving us an opportunity to revise our manuscript, we appreciate Reviewer 1 very much for positive and constructive comments and suggestions on our manuscript entitled “In-situ nano-SiO2 electrospun polyethylene oxide based nano-fiber composite solid polymer electrolyte for high performance lithium-ion batteries”.

    We have studied comments carefully and have made revision which marked in red in the paper. We have tried our best to revise our manuscript according to the comments. Please refer to the attachment for the specific revisions.

Reviewer 2 Report

It is a a very interesting paper showing how SiO2 nanoparticles could affect the crystallinity of PEO and consequently his ionic conductivity.

The paper could shighly be improved :

2.3 and figure 2, the SEM images were recorded with with detector and at what acceleration voltage? I suppose that the voltage is quite low and the detector is a  Secondary  Electron detector . It would be nice to see the Backscattered Electrons images to nicely observe the SiO2 particles within the PEO . 

The EDS probe size in a SEM is around 1 µm3. I do not understant how it could be usefull to observe 45nm or less SiO2 particles. At what voltage was recorded the EDS mappinfs ?

On the Fig2 e, we can see an histogramn of size repartition. How many particles were measured to obtain this hIstogram. What was the resolution of the images (error) ?

2.3

Authors present XRD peaks in 2theta, but I do not see what was the wavelenght used to record the patterns.

On figure 3c, auhtors do not talk about the origin of the peaks quite intense at 65 anf 80 °.

Could the authors explain clearly the sentence :

" The increase in peak intensity from 2.=19.2° (120) can be achieved, signifying that in situ SiO2 nanospheres interact strongly with PEO chain"segments as above FTIR result mentioned.

How the interaction of SiO2 with PEO chain could affect the intensity of only one peak?

Author Response

Dear Reviewer:

     We are grateful to Reviewer 2 for effort reviewing our paper and positive feedback. The summary of our work as written by the reviewer is precise. Here below we address the questions and suggestions raised by the Reviewer 2. Please refer to the attachment for the specific revision.

Round 2

Reviewer 1 Report

The authors addressed the points raised by the referee and made the changes in the revised version of the paper. Under these considerations, I think that the paper can be accepted for publication in “Nanomaterials” (Recommendation: accept).